# PRUNING IN TRAINING: LEARNING AND RANKING SPARSE CONNECTIONS IN DEEP CONVOLUTIONAL NETWORKS

## ABSTRACT

This paper proposes a Pruning in Training (PiT) framework of learning to reduce the parameter size of networks. Different from existing works, our PiT framework employs the sparse penalties to train networks and thus help rank the importance of weights and filters. Our PiT algorithms can directly prune the network without any fine-tuning. The pruned networks can still achieve comparable performance to the original networks. In particular, we introduce the (Group) Lasso-type Penalty (L-P /GL-P), and can alternatively implement them using (Group) Split LBI (S-P / GS-P)– an regularization solution path with corresponding penalties to regularize the networks, and a pruning strategy proposed by Fu et al. (2016b) is used to help prune the network. We conduct the extensive experiments on MNIST, Cifar-10, and *mini*ImageNet. The results validate the efficacy of our proposed methods. Remarkably, on MNIST dataset, our PiT framework can produce a small network which only has 17.5% parameter size of LeNet-5, and achieves the 98.47% recognition accuracy.

## 1 INTRODUCTION

The expressive power of Deep Convolutional Neural Networks (DNNs) comes from the millions of parameters, which are optimized by various algorithms such as Stochastic Gradient Descent (SGD), and Adam Kingma & Ba (2015). However, one has to strike a trade-off between the representation capability and computational cost, caused by the plenty of parameters in the real world applications, *e.g.,* robotics, self-driving cars, and augmented reality. Pruning significant number of parameters would be essential to reduce the computational complexity and thus facilitate a timely and efficient fashion on a resource-limited platform, *e.g.* devices of Internet of Things (IoT). In addition, it has long been conjectured that the state-of-the-art DNNs may be too complicated for most specific tasks; and we may have the free lunch of "reducing $2\times$ connections without losing accuracy and without retraining" Han et al. (2015b).

To compress DNNs, recent efforts had been made on learning the DNNs of small size. They either reduce the number and size of weights of parameters of original networks, and fine-tune the pruned networks Abbasi-Asl & Yu (2017); Yang et al. (2018), or distill the knowledge of large model Hinton et al. (2014), or directly learning the compact and lightweight small DNNs, such as ShuffleNet Ma et al. (2018), MobileNet Howard et al. (2017), and SqueezeNet Iandola et al. (2017). Note that, (1) to efficiently learn the compressed DNNs, previous works had to introduce additional computational cost in fine-tuning, or training the updated networks; (2) it is not practical nor desirable to learn the *tailored*, or *bespoke* networks for any applications, beyond computer vision tasks.

To this end, the center idea of this paper is to propose a Pruning in Training (PiT) framework that enables pruning networks in the training process. Particularly, the sparsity regularizers, including *lasso-type*, and *split LBI penalties* are applied to train the networks. Such regularizers not only encourage the sparsity of DNNs, *i.e.*, fewer (sparse) connections with non-zero values, but also can accelerate the speed of DNNs convergence. Furthermore, in the learning process, we can iteratively compute the regularization path of layer-wise parameters of DNNs. The parameters can be ranked by the regularization path in a descending order, as Fu et al. (2016a). The parameters in the high rank are in the high priority of not being pruned.

More importantly, our PiT can learn the sparse structures of DNNs, and utilize the functionality of filters and connection weights (in fully connected layers). In the optimal cases, the weights (or filters) of each layer should be learned fully orthogonal to each other and thus formulate an orthogonal basis. The orthogonal constraint may be only enforced as the initialization (*e.g.*, SVD Jia (2017) and Saxe et al. (2014)), or via the other regularization tricks, such as dropout preventing co-adaption Srivastava et al. (2014), or batch normalization reducing the internal covariate shift of hidden layers Ioffe & Szegedy (2015). Therefore, our PiT can help uncover redundant information in a network by compressing less important filters and weights, and facilitate pruning out more interpretable networks.

## 2 RELATED WORKS

The deeper and wider deep CNN architectures can enable the superior performance on various tasks, and yet cause the prohibitively expensive computation cost. To efficiently train the networks, the regularization is usually applied to the weight parameters (Sec. 2.1). It is also essential to prune networks to reduce the size of networks (Sec. 2.2)

### 2.1 NETWORK REGULARIZATION

Due to large number of parameters, the deep networks require large amount of memory and computational resources, and are inclined to overfit the training data. To alleviate this problem, it is essential to regularize the networks in training stage; such as dropout Srivastava et al. (2014) preventing the co-adaptation, and adding $L_2$ or $L_1$ regularization to weights. In particular, the $L_1$ regularization enforces the sparsity on the weights and results in a compact, memory-efficient network with slightly sacrificing the prediction performance Collins & Kohli (2014). Further, group sparsity regularization Yuan & Lin (2006) can also been applied to deep networks with desirable properties. Alvarez *et al.* Alvarez & Salzmann (2016) utilized a group sparsity regularizer to automatically decide the optimal number of neuron groups. The structured sparsity Wen et al. (2016a); Yoon & Hwang (2017) has also been investigated to exert good data locality and group sparsity. Different from these works, the (Group) Split LBI penalty is for the first time, introduced to regularize the networks. This regularization term can not only enforce the structured sparsity, but also can efficiently compute the solution paths of each variable.

### 2.2 NETWORK PRUNING

Compressing the networks involves the pruning and compressing the weights and filters of DNNs. The common strategies include (1) *matrix decomposition methods* Jaderberg et al. (2014); Zhang et al. (2016; 2015); Tai et al. (2016) by decomposing the weight matrix of DNNs as a low-rank product of two smaller matrices; (2) *low-precision weights methods* Zhu et al. (2017); Zhou et al. (2017) by learning to store low-precision weights of DNNs; and (3) *pruning methods* Han et al. (2015b); Li et al. (2017) directly removing weights of connections, or neurons.

Our framework is one of pruning methods. Previous pruning works, iteratively prune the weights or neurons, and fine-tune the network Han et al. (2015b); Guo et al. (2016). Remarkably, network regularization is of significant important in pruning methods. The sparse properties of features maps and/or weights of DNNs exerted by network regularization, are utilized in Wen et al. (2016b); Lebedev & Lempitsky (2016). Luo *et al.* Luo et al. (2017) adopt the statistics information from next layer to guide and save the importance of filters of the current layer. Molchanov et al. Molchanov et al. (2017) employed Taylor expansion to approximate the change of cost function which can be further utilized as the criterion in pruning network parameters. A LASSO-based channel selection strategy is investigated in He et al. (2017). Abbasi-Asl *et al.* Abbasi-Asl & Yu (2017) defined a filter importance index of greedy pruning the network. Comparing with all the methods, our framework is different in two points: (1) *Criterion of importance of weights and filters*. We rank the importance of weights and filters by their solution paths computed by sparse regularizers, rather than designing the elaborated metrics as previous works Abbasi-Asl & Yu (2017); Yang et al. (2018). Specifically, our algorithm is a process of solving the discrete partial differential equations; and our framework can result in the solution paths of optimizing the weights and filters, whose importance are ranked,

according to the selected order in the path, as Fu et al. (2016b). (2) *Pruning in training*: once DNNs are trained, we simply prune out less important weights/filter by a threshold.

# 3 METHODOLOGY

In this section, the Residual Network (ResNet) structureHe et al. (2016) is employed to elaborate our framework. Our algorithms can be used in the other DNNs, *e.g.* Lenet-5.

## 3.1 NOTATION

We adopt the notations of ResNet strucutre, in which the output of the $i$th block $\mathcal{O}_i$ can be represented as:

$$\mathcal{O}_i = \mathcal{F}\left(x, \{W\}_i\right) + \mathcal{W}_i x$$

where $x$ the input of the first layer of the $i$th block, $\{W\}_i$ and $\mathcal{W}_i$ respresent the filter weights in the $i$th block and the shortcut weight matrix, respectively. The function $\mathcal{F}(\cdot)$ represent the multiple convolutional layers. Denote the weight matrix of the first convolutional layer as $W_{conv1}$ and that of the fully connected layer as $W_{fc}$. Suppose there are $I$ blocks, then we denote all the parameters of the network as $\Theta := \{W_{conv1}, \{W\}_1, ..., \{W\}_I, \mathcal{W}_1, ..., \mathcal{W}_I, W_{fc}\}$ and $\Theta_{-W} := \Theta \backslash W$ for $W \in \Theta$. Our key objective is to train a sparse DNN of less parameters, and yet comparable performance to the non-sparse DNN. The training function of DNN is defined as,

$$\min_{\Theta} L\left(\Theta; \mathcal{X}, y\right) + \lambda \cdot P\left(\Theta\right) \tag{1}$$

where $P\left(\cdot\right)$ is the penalty function of parameters $\Theta$. If we use $(\mathcal{X}, y)$ as the sample set of the dataset; then in classification task, the loss function is the cross-entropy function as

$$L(\Theta; \mathcal{X}, y) = -\frac{1}{N} \sum_{n=1}^{N} \sum_{k=1}^{K} y_{n,k} \log\{p_{n,k}(\Theta)\}, \tag{2}$$

where $N$ and $K$ are the number of samples and classes and $p_{n,k}\left(\Theta\right)$ denotes the probability of the $n$th sample belongs to class $k$. Generally, we can use Stochastic Gradient Descent (SGD) algorithm to update $\Theta$; and the algorithm is summarized in algorithm 1.

---

**Algorithm 1** SGD for ResNet

---

1: **Input:** Learning rate $\eta$, $\mathcal{X}$ and $y$
2: **Initialize:** $k = 0$, $\Theta^k$ is initialized randomly.
3: **Iteration**
4: $\quad \Theta^{k+1} = \Theta^k - \eta \nabla_\Theta L(\Theta^k)$
5: **Output:** $\{\Theta^{k+1}\}$

---

## 3.2 REGULARIZATION ON ONE LAYER

One direct intuition is to adopt the sparsity regularization on the parameters, or those of the one layer of the network, such as Liu et al. (2015); Wen et al. (2016b). To reduce the number of connection weights, one can consider different types of regularization, including (1) Lasso-type penalty ($L_1$), (2) Group-Lasso-type penalty Yuan & Lin (2006); (3) An iterative regularization path with structural sparsity (*e.g.*, elastic net Zou & Hastie (2005), and Split LBI Huang et al. (2016)): here we employ the Split LBI which learns the structural sparsity via variable splitting and Linearized Bregman Iteration (LBI), due to the computational efficiency of the LBI, and model selection consistency,

**Lasso-type penalty** can be directly implemented on the fully connection layer $i$ as,

$$P(W) = \|W\|_1; \tag{3}$$

**Group-Lasso-type penalty** Yuan & Lin (2006) aims at regularizing the groups of parameters $\Theta$, and $W^{(g)}$ is a group of partial weights in $\Theta$,

$$P(W) = \sum_{g=1}^{G} \|W^{(g)}\|_2 \tag{4}$$

where $\|W^{(g)}\|_2 = \sqrt{\sum_{i=1}^{|W^{(g)}|} \left(W_i^{(g)}\right)^2}$, and $|W^{(g)}|$ is the number of weights in $W^{(g)}$; $G$ is the total number of groups.

### 3.3 OPTIMIZATION

This Split LBI algorithm Huang et al. (2016) introduces an augmented variable $\Gamma$ which is enforced sparsity and kept close to $W$, by variable splitting term $\frac{1}{2\nu}\|\Gamma - W\|_2^2$. Then the objective function turns to:

$$L(W, \Gamma; \Theta_{-W}, \mathcal{X}, y) = L(W; \Theta_{-W}, \mathcal{X}, y) + \frac{1}{2\nu}\|\Gamma - W\|_2^2, (\nu > 0)$$

To enforce the sparsity of $\Gamma$, we here implement the LBI algorithm on the $W$, and the algorithm can be summarized in algorithm 2, where

$$\text{Prox}_L(Z) = \arg\min_W \frac{1}{2}\|W - Z\|_2^2 + P(W) \tag{5}$$

The 5th-8th lines are Split LBI algorithm, which returns a regularization path of $\{\Theta_{-W}^k, W^k, \widetilde{W}^k, \Gamma^k\}$. It starts from the null model with $\Gamma^0 = 0$, and tends to select more and more variables as the algorithm evolves, until over-fitted. At each step, the sparse estimator $\widetilde{W}^k$ is the projection of $W^k$ onto the subset of the support set of $\Gamma^k$. The remainder of the projection is affected by weak signals with small magnitude and mostly the ones mainly affected by random noise. Particularly, we highlight several points,

- The $\kappa$ is the damping factor, which enjoys the low bias with larger value, however, at the sacrifice of high computational cost. The $\alpha$ is the step size. In Huang et al. (2016), it has been proved that the $\alpha$ is the inverse scale with $\kappa$ and should be small enough to ensure the statistical property. In our scenario, we set it to $0.01/\kappa$.

- The $t^k = k\alpha$ is the regularization parameter, which plays the similar role with $\lambda$ in Lasso. It's the trade-off between underfiting and overfiting, which can be determined via the loss/accuracy on the validation dataset.

- The $\nu$ controls the difference between $\widetilde{W}$ and $W$. In Huang et al. (2016), it has been proved that larger value of $\nu$ can enjoy better model selection consistency, however may suffer from the larger parameter estimation error. In Sun et al. (2017); Zhao et al. (2018), it has been proved that as long as $\nu \nrightarrow 0$, the dense estimator $W$ can enjoy better prediction error by leveraging weak signals. We will discuss it in the next subsection.

- Each component of the closed form solution $W \in \mathbb{R}^{p_1 \times p_2}$ in equation 5 can be simplified as,

$$\begin{cases} \kappa \max(0, 1 - 1/\|Z^{(g)}\|_2)Z^{(g)} & \text{P defined in equation 4 for g} \in \{1, ..., G\} \\ \kappa \max(0, 1 - 1/|Z_i|)Z_i & \text{P defined in equation 3 for i} \in \{1, ..., p_1 \times p_2\} \end{cases} \tag{6}$$

### 3.4 PRUNING STRATEGY

The pruning algorithm is inspired by the Fu et al. (2016b). Particularly, it has been pointed out in Zhao et al. (2018) that the dense estimator can be orthogonally decomposed into three parts: strong signals which correspond to non-zero elements in $\widetilde{W}$, weak signals and random noise. Due to the ability to leverage additional weak signals as long as $\nu$ is large enough, it has been proved theoretically and experimentally that, the dense estimator outperforms the sparse estimator in prediction.

---

**Algorithm 2** SGD for ResNet with Split LBI

---

1: **Input:** Learning rate $\eta$, $\nu > 0$, step size of LBI $\alpha$, damping factor $\kappa > 0$, $\mathcal{X}$ and $y$
2: **Initialize:** $k = 0$, $\Theta^k$ is initialized randomly, $\Gamma^k = Z^k = 0$
3: **Iteration**
4:    $\Theta^{k+1}_{-W} = \Theta^k_{-W} - \eta \nabla_{\Theta_{-W}} L(\Theta^k_{-W}, W^k, \Gamma^k)$
   *# LBI update*
5:    $\boldsymbol{W^{k+1} = W^k - \kappa\alpha\nabla_W L(\Theta^k_{-W}, W^k, \Gamma^k)}$
6:    $\boldsymbol{Z^{k+1} = Z^k - \alpha\nabla_\Gamma L(\Theta^k_{-W}, W^k, \Gamma^k)}$
7:    $\boldsymbol{\Gamma^{k+1} = \kappa\mathbf{Prox}_J(Z^{k+1})}$
8:    $\boldsymbol{\widetilde{W}^{k+1} = W^k \circ \left[\mathbf{1}\{i \in S^{k+1}\}\right]_{i,j} \left(S^{k+1} = \mathrm{supp}(\Gamma^k)\right)}$
9: **Output:** $\{\Theta^{k+1}_{-W}, W^{k+1}, \widetilde{W}^{k+1}, \Gamma^{k+1}\}$

---

This inspires us to sequentially consider all available solutions for all sparse variables along the Regularization Path (RP) by gradually decreasing the values of regularization coefficients. Specifically, we can order the parameter set $\Theta$ according to the magnitude values of weights $W$. Following this order, we identify the top $r\%$ of weights in $\Theta_r$. The complementary set $\Theta_{1-r} = \Theta \backslash \Theta_r$ can be pruned. Compared to the pruning methods in Han et al. (2015a), we can prune the weights in the training process and do not need to fine-tune the weights.

### 3.5 REGULARIZATION ON MULTIPLE LAYERS

Furthermore, one can easily extend algorithm 2 to prune at $L$ $(L > 1)$ layers. We take the Split LBI as an example; the other two methods can also be directly applied to multiple layers. The corresponding algorithm is described in algorithm 3.

## 4 EXPERIMENTS

---

**Algorithm 3** SGD for ResNet with Split LBI on multiple layers

---

1: **Input:** Learning rate $\eta$, $\nu > 0$, step size of LBI $\alpha$, damping factor $\kappa > 0$, $\mathcal{X}$ and $y$
2: **Initialize:** $k = 0$, $\Theta^k$ is initialized randomly, $\Gamma^k_1 = Z^k_1 = 0$, ..., $\Gamma^k_L = Z^k_L = 0$
3: **Iteration**
4:    $\Theta^{k+1}_{-W} = \Theta_{-\{W^k_1,...,W^k_L\}} - \eta\nabla_{\Theta_{-\{W^k_1,...,W^k_L\}}} L(\Theta_{-\{W^k_1,...,W^k_L\}}, \{W^k_1,...,W^k_L\}, \{\Gamma^k_1,...,\Gamma^k_L\})$
   *# LBI update at L layers*
5:    **For** $l = 1, ..., L$
6:       $\boldsymbol{W^{k+1}_l = W^k_l - \kappa\alpha\nabla_W L(\Theta_{-\{W^k_1,...,W^k_L\}}, \{W^k_1,...,W^k_L\}, \{\Gamma^k_1,...,\Gamma^k_L\})}$
7:       $\boldsymbol{Z^{k+1}_l = Z^k_l - \alpha\nabla_\Gamma L(\Theta_{-\{W^k_1,...,W^k_L\}}, \{W^k_1,...,W^k_L\}, \{\Gamma^k_1,...,\Gamma^k_L\})}$
8:       $\boldsymbol{\Gamma^{k+1}_l = \kappa\mathbf{Prox}_J(Z^{k+1}_l)}$
9:       $\boldsymbol{\widetilde{W}^{k+1}_l = W^k_l \circ \left[\mathbf{1}\{i \in S^{k+1}_l\}\right]_{i,j} \left(S^{k+1}_l = \mathrm{supp}(\Gamma^k_l)\right)}$
10:   **End**
11: **Output:** $\{\Theta_{-\{W^{k+1}_1,...,W^{k+1}_L\}}, \{W^{k+1}_1,...,W^{k+1}_L\}, \{\widetilde{W}^{k+1}_1,...,\widetilde{W}^{k+1}_L\}, \{\Gamma^{k+1}_1,...,\Gamma^{k+1}_L\}\}$

---

We conduct the experiments on three datasets, *namely*, MNIST, CIFAR10, and *Mini*ImageNet. We use the standard supervised training and testing splits on all datasets, except *Mini*ImageNet, whose setting is splitted by ourselves, and will be released. The classification accuracy is reported on each dataset.

**Competitors**. We compare three methods of pruning networks. (1) *Plain*: we train a plain network and use the $L_2-$ penalty $P(W) = \|W\|_2$. For all layers, we set the coefficient $\lambda$ as $5e-4$ in Eq (1). We prune the trained network by ranking the weights and filters, in term of their magnitude values in the descending order. This pruning strategy can be taken as a simplified version of our pruning algorithm in Sec. 3.4. (2) *Rand*. We randomly remove the weights or filters in the networks. This is a naive baseline. (3) *Ridge-Penalty* (*R-P*) Han et al. (2015b): We use the same ranking methodology

| (%) | 100 | 25 | 12.5 | 6.25 | 3.13 | 1.57 |
|---|---|---|---|---|---|---|
| Plain | 99.17 | 60.87 | 29.65 | 20.82 | 20.82 | 20.82 |
| Rand | 99.11 | 43.72 | 30.07 | 18.35 | 24.12 | 22.62 |
| R-P | 99.12 | 61.05 | 46.91 | 30.34 | 30.34 | 30.34 |
| GL-P | 99.05 | 74.29 | 47.28 | 28.58 | 28.58 | 28.58 |
| GS-P | 99.00 | 85.09 | 32.58 | 22.88 | 22.88 | 22.88 |

(1) Pruning the conv.c3 layer

| (%) | 100 | 25 | 12.5 | 6.25 | 3.13 | 1.57 |
|---|---|---|---|---|---|---|
| Plain | 99.12 | 80.46 | 62.61 | 45.49 | 32.34 | 21.30 |
| Rand | 99.19 | 62.23 | 37.71 | 23.58 | 18.58 | 14.36 |
| R-P | 99.16 | 75.47 | 60.31 | 37.97 | 26.11 | 18.11 |
| GL-P | 98.95 | 98.95 | 90.29 | 60.37 | 32.91 | 20.31 |
| GS-P | 98.97 | 98.96 | *98.67* | 68.27 | 42.10 | 24.95 |

(2) Pruning the conv.c5 layer

| (%) | 100 | 25 | 12.5 | 6.25 | 3.13 | 1.57 |
|---|---|---|---|---|---|---|
| Plain | 99.10 | 96.73 | 95.65 | 89.60 | 78.40 | 64.17 |
| Rand | 99.09 | 91.56 | 71.05 | 51.92 | 33.65 | 29.91 |
| R-P | 99.13 | 96.39 | 95.31 | 91.29 | 82.75 | 68.35 |
| L-P | 99.10 | 98.89 | 98.89 | 98.89 | 98.89 | 98.89 |
| S-P | 99.13 | 98.73 | 98.61 | 98.23 | 96.75 | 92.53 |

(3) Pruning the fc.c6 layer

| (%) | 100 | 25 | 12.5 | 6.25 | 3.13 | 1.57 |
|---|---|---|---|---|---|---|
| Plain | 99.11 | 98.36 | 94.52 | 78.15 | 68.72 | 47.35 |
| Rand | 99.15 | 66.29 | 50.76 | 34.61 | 24.05 | 19.78 |
| R-P | 99.13 | 98.62 | 96.50 | 84.04 | 67.08 | 56.34 |
| L-P | 99.10 | 99.11 | 99.11 | 99.11 | 99.09 | 96.47 |
| S-P | 99.03 | 99.00 | 99.00 | 99.01 | 99.03 | 95.41 |

(4) Pruning the fc.f7 layer

Table 1: Pruning one layer in LeNet-5 on MNIST dataset (Top-1 Accuracy).

to rank the weights and filters by $L_2$ regularization path. For that particular layer that we want to do the pruning, the coefficient $\lambda$ would be finally set as $1e-3$.

We also compare two variants of our PiT framework. (4) *Lasso-type penalty* or *Group-Lasso-type penalty* (*L-P / GL-P*): the L-P is used to prune the weights of fully connected layers, and we employ the GL-P to directly remove the filters of convolutional layers. (5) *Split LBI* or *Group Split LBI penalty*(*S-P / GS-P*): the split BLI penalty is utilized to prune the weights. Accordingly, we have the Group Split LBI penalty by regularizing the groups of filter parameters as Yuan & Lin (2006). Note that all the results are trained for one time; and we do not have fine-tuning step after the pruning.

### 4.1 LETNET ON MNIST

The handwritten digits MNIST dataset is widely used to experimentally evaluate the machine learning methods. We use the standard supervised split and LeNet-5 LeCun et al. (1998) which is composed of 3 convolutional layers and 2 fully connected layers. All the models are trained and get converged in 50 epochs. Note that each experiment is repeated for five times, and the averaged results are reported. In the experiments, we consider saving the portion of 100%, 50%, 25%, 12.5%, 6.25%, 3.13%, and 1.57% of original parameters on each layer. Please refer to the Appendix for more detailed results.

**Pruning each layer**. The results are shown in Tab. 1. We employ our PiT algorithms to prune each individual layers of LeNet-5, while we keep the parameters of the other layers unchanged. We have the following observations:

(1) On two fully connected layers (fc.f6 and fc.f7), both the L-P and S-P of our PiT framework work very well. For example, on the fc.f7 layer, our S-P only has 1.57% of the parameters on these layers. Surprisingly, our performance is only 0.03% lower than that of the original network. In contrast, we compare the pruning results with the baseline: Plain, Rand, and R-P. There is significant performance dropping with the more parameters pruned. This shows the efficacy of our PiT framework.

(2) On the convolutional layer (conv.c5), our L-P and S-P layers also achieve remarkable results. Note that the conv.c5 layer has $48k$ out of $60k$ number of parameters in Lenet-5. We show that our S-P saves 12.5% of total parameters of this layer (*i.e.*, $42k$ number of parameters have been removed on this layer) and the results get only dropped by 0.3%. This demonstrates that our PiT framework indeed can save the relatively important weights and filters, and effectively do the network pruning.

(3) The conv.c3 layer is another convolutional layer in LeNet-5. We found that this layer is very important to maintain a good performance of overall network. Nevertheless, the results of our pruning L-P and S-P are still better than the other baselines.

**Pruning two layers.** Totally, the LeNet-5 has $60k$ parameters, while the *conv.c5* and *fc.f6* have $48k$ and $10k$ number of parameters respectively. That means these two layers have the most number of

| | (%) | 100 | 25 | 12.5 | 6.25 | 3.13 | 1.57 |
|---|---|---|---|---|---|---|---|
| | Plain | 99.03 | 33.72 | 46.29 | 42.29 | 57.24 | 33.83 |
| | Rand | 99.02 | 37.25 | 25.84 | 17.13 | 11.09 | 11.27 |
| fc.f6 + fc.f7 | R-P | 99.14 | 67.35 | 44.32 | 66.56 | 45.14 | 31.83 |
| | L-P | 99.10 | 98.58 | 98.58 | 98.58 | 98.58 | 98.60 |
| | S-P | 99.05 | 98.71 | 98.71 | 98.71 | 98.68 | 98.32 |
| Com-Rat(%) | | — | 91.83 | 87.74 | 85.70 | 84.68 | 84.17 | 83.91 |
| | (%) | 100 | 25 | 12.5 | 6.25 | 3.13 | 1.57 |
| | Plain | 99.16 | 70.04 | 51.58 | 32.83 | 14.54 | 20.84 |
| | Rand | 99.16 | 40.24 | 22.76 | 15.30 | 12.06 | 10.36 |
| conv.c5+fc.f6 | R-P | 99.08 | 83.21 | 50.68 | 34.07 | 25.39 | 12.87 |
| | GL-P / L-P | 98.96 | 97.92 | 97.92 | 73.44 | 28.08 | 16.84 |
| | GS-P / S-P | 98.69 | 98.47 | *98.47* | 88.77 | 50.71 | 28.14 |
| Com-Rat(%) | | — | 52.91 | 29.37 | 17.60 | 11.71 | 8.77 | 7.30 |

Table 2: Pruning two layers in LeNet-5 on the MNIST dataset. Each column indicates the percentage of parameter saved on these two layers. *Com-Rat*, is short for the compression ratio of the total network, *i.e.*, the ratio of saved parameters divided the total number of parameters of LeNet-5.

| (%) | 100 | 25 | 12.5 | 1.57 | (%) | 100 | 25 | 12.5 | 1.57 | (%) | 100 | 25 | 12.5 | 1.57 |
|---|---|---|---|---|---|---|---|---|---|---|---|---|---|---|
| Plain | 92.96 | 28.91 | 29.21 | 11.76 | Plain | 93.44 | 37.32 | 20.60 | 18.84 | Plain | 93.42 | 41.69 | 18.36 | 27.28 |
| Rand | 93.58 | 17.04 | 15.14 | 13.97 | Rand | 92.90 | 25.63 | 30.82 | 34.33 | Rand | 93.44 | 14.39 | 13.42 | 11.43 |
| R-P | 93.75 | 82.35 | 55.53 | 21.19 | R-P | 93.95 | 64.03 | 46.21 | 27.67 | R-P | 93.35 | 54.27 | 32.17 | 32.51 |
| GL-P | 93.54 | 93.53 | 93.30 | 89.20 | GL-P | 93.57 | 93.57 | 93.57 | 93.57 | GL-P | 93.60 | 93.62 | 93.58 | 93.61 |
| GS-P | 93.17 | 93.23 | 93.27 | 93.27 | GS-P | 93.52 | 93.53 | 93.53 | 93.49 | GS-P | 93.27 | 93.18 | 93.26 | 93.24 |
| (1) Pruning Block#1.0 | | | | | (2) Pruning the Block#1.1 | | | | | (3) Pruning the Block#2.0 | | | | |
| (%) | 100 | 25 | 12.5 | 1.57 | (%) | 100 | 25 | 12.5 | 1.57 | (%) | 100 | 25 | 12.5 | 1.57 |
| Plain | 93.48 | 49.77 | 40.31 | 36.75 | Plain | 93.41 | 55.48 | 22.94 | 35.36 | Plain | 93.11 | 51.73 | 34.17 | 18.86 |
| Rand | 93.50 | 31.60 | 35.15 | 37.40 | Rand | 93.11 | 37.42 | 36.66 | 37.10 | Rand | 93.84 | 46.47 | 46.62 | 43.55 |
| R-P | 93.78 | 64.74 | 51.29 | 49.65 | R-P | 93.61 | 74.24 | 57.27 | 38.32 | R-P | 93.73 | 54.31 | 67.49 | 56.04 |
| GL-P | 93.50 | 93.50 | 93.50 | 93.51 | GL-P | 93.66 | 93.66 | 93.67 | 93.61 | GL-P | 93.54 | 93.54 | 93.54 | 93.54 |
| GS-P | 93.50 | 93.16 | 93.27 | 93.29 | GS-P | 93.59 | 93.63 | 93.48 | 93.42 | GS-P | 93.82 | 93.46 | 93.33 | 93.26 |
| (4) Pruning the Block#2.1 | | | | | (5) Pruning the Block#3.0 | | | | | (6) Pruning the Block#3.1 | | | | |
| (%) | 100 | 25 | 12.5 | 1.57 | (%) | 100 | 25 | 12.5 | 1.57 | | | | | |
| Plain | 93.08 | 82.70 | 54.18 | 38.29 | Plain | 93.37 | 90.01 | 85.29 | 75.53 | | | | | |
| Rand | 93.44 | 48.88 | 40.89 | 37.38 | Rand | 93.57 | 82.86 | 79.16 | 76.73 | | | | | |
| R-P | 93.55 | 88.63 | 69.91 | 47.40 | R-P | 93.63 | 88.20 | 87.16 | 70.77 | | | | | |
| GL-P | 93.65 | 93.65 | 93.65 | 93.68 | GL-P | 93.79 | 93.79 | 93.79 | 93.77 | | | | | |
| GS-P | 93.61 | 93.50 | 93.45 | 93.41 | GS-P | 93.61 | 93.93 | 93.83 | 93.89 | | | | | |
| (7) Pruning the Block#4.0 | | | | | (8) Pruning the Block#4.1 | | | | | | | | | |

Table 3: Pruning each block in ResNet-18 on Cifar-10 dataset. Note that each block has two CNN layers.

parameters. In this case, we utilize our PiT algorithms to prune both *fc.f6 + fc.f7*, and *conv.c5+fc.f6* layers. The results are reported in Tab. 2. We can show that our PiT framework can still efficiently compress the network while preserve significant performance.

**The best compressed model.** When we prune the *conv.c5* and *fc.f6* layers, our model can achieve the best and efficient performance. With only 17.60% parameter size of original LeNet-5, our model can beat the performance as high as 98.47%. Remarkably, our PiT framework has not done any fine-tuning and re-training the pruned network by any other dataset. This suggests that our PiT can indeed uncover the important weights and filters. Our best models will be downloaded online.

| Block | (%) | 100 | 25 | 12.5 | 6.25 | 3.13 | 1.57 |
|---|---|---|---|---|---|---|---|
|  | Plain | 93.13 | 75.87 | 20.64 | 10.01 | 10.00 | 10.00 |
|  | Rand | 93.28 | 11.82 | 10.28 | 10.08 | 10.12 | 10.15 |
| #4.0 + #4.1 | R-P | 93.46 | 78.00 | 27.48 | 10.02 | 10.00 | 10.00 |
|  | GL-P | 94.03 | 93.96 | 93.96 | 93.92 | 93.20 | 92.11 |
|  | GS-P | 92.90 | 92.92 | 92.88 | 92.93 | 92.85 | 92.70 |
| Com-Rat(%) | — | | 62.29 | 43.44 | 34.02 | 29.30 | 26.95 | 25.77 |
| **Block** | **(%)** | **100** | **25** | **12.5** | **6.25** | **3.13** | **1.57** |
|  | Plain | 93.13 | 89.22 | 51.70 | 10.68 | 10.00 | 10.00 |
|  | Rand | 92.94 | 12.55 | 10.00 | 10.00 | 10.00 | 10.00 |
| #3.1 + #4.0+#4.1 | R-P | 93.78 | 46.10 | 14.67 | 10.01 | 16.05 | 10.43 |
|  | GL-P | 92.65 | 92.64 | 92.47 | 92.17 | 91.43 | 90.96 |
|  | GS-P | 91.55 | 91.54 | 91.69 | 91.07 | 90.95 | 91.29 |
| Com-Rat(%) | — | | 56.91 | 35.36 | 24.59 | 19.20 | 16.51 | 15.16 |
| **Block** | **(%)** | **100** | **25** | **12.5** | **6.25** | **3.13** | **1.57** |
|  | Plain | 92.97 | 10.00 | 10.00 | 10.00 | 10.00 | 10.00 |
|  | Rand | 92.83 | 13.94 | 12.67 | 14.72 | 12.57 | 13.64 |
| #3.0+#3.1+#4.0+#4.1 | R-P | 93.96 | 18.63 | 10.00 | 10.00 | 10.00 | 10.00 |
|  | GL-P | 91.15 | 91.15 | 90.18 | 87.33 | 81.82 | 73.77 |
|  | GS-P | 90.61 | 89.19 | 88.47 | 88.26 | 88.40 | *87.94* |
| Com-Rat(%) | — | | 52.88 | 29.32 | 17.53 | 11.64 | 8.70 | 7.23 |

Table 4: Pruning multiple blocks in ResNet-18 on Cifar-10 dataset. (Chance-level = 10%). Com-Rat, is short for compression ratio of the total network, as in Tab. 2.

## 4.2 RESNET-18 ON CIFAR10 DATASET

The CIFAR-10 dataset consists of 60,000 images of size $32 \times 32$ in 10 classes, with 6000 images per class on average. There are 50,000 training images and 10,000 test images. We use the standard supervised split; and ResNet-18 is employed as the classification network. All the models are trained and get converged in 40 epochs. Note that each experiment is repeated for five times, and the averaged results are reported. We still show the results which have 100%, 50%, 25%, 12.5%, 6.25%, 3.13%, and 1.57% parameter size of original networks on each layer.

**Pruning one Residual Block**. The results are shown in Tab. 3. In this table, we apply our PiT algorithm on one residual block while the other layers are unchanged. We draw several conclusions,

(1) Our PiT framework (*i.e.*, GS-P and GL-P) can efficiently train and prune the network. From Block #3.0 – Block #4.1, surprisingly the pruned network with 1.57% of original parameter size of ResNet-18, can also achieve almost the same recognition accuracies as the non-pruned ResNet-18. From Block #1.0 – Block #2.1, the smallest pruned ratio of PiT can still hit significant high performance if compared with the other competitors. This reflects the efficacy of our pruning algorithm. In particular, in the training process, our PiT framework is optimized to learn and select the important weights or filters; and our PiT can thus conduct a direct dimension reduction of these parameters.

(2) By the increased ratio of pruned parameters, the R-P method can also have better performance than Rand, and Plain methods. This shows that our pruning algorithm also works in the general cases. However, the R-P is not enforcing the sparse constraints in learning the weight parameters of network. Thus it has inferior performance to two PiT methods.

**Pruning multiple blocks.** The ResNet-18 totally has around $10.95M$ parameters. Block #4.1, #4.0, #3.1, #3.0 have $4.7M$, $3.5M$, $1.2M$, and $0.88M$ parameters. These blocks have the most number of parameters; and we prune these multiple blocks. The results are shown in Tab. 4. Note that even only 1.57% parameter size of those layers are saved, our PiT algorithms (GL-P, and GS-P) can still remain remarkable high recognition accuracy. Again it shows the efficacy of our PiT framework.

| (%) | 100 | 50 | 25 | 12.5 | 6.25 | 3.13 | 1.57 |
|---|---|---|---|---|---|---|---|
| Plain | 83.92 | 13.53 | 8.12 | 5.32 | 5.29 | 5.92 | 6.31 |
| Rand | 82.36 | 17.90 | 6.52 | 6.38 | 7.90 | 9.58 | 8.67 |
| R-P | 82.79 | 13.72 | 7.10 | 6.38 | 6.29 | 6.52 | 5.76 |
| L-P / GL-P | 81.09 | 81.09 | 76.43 | 75.06 | 68.42 | 55.25 | 33.49 |
| S-P / GS-P | 78.95 | 77.81 | 73.92 | 70.65 | 68.67 | 67.58 | *65.17* |

Table 5: Top 5 accuracy on *mini*Imagenet by pruning ResNet-18, the fully connected layer, Block#4.0 and #4.1 layers.

### 4.3 ResNet-18 on *mini*ImageNet dataset

The *mini*ImageNet dataset is a subset of ImageNet and is composed of 60,000 images in 100 categories. In each category, we take 500 images as training set and other 100 as testing set. We also use the ResNet-18 structure on *mini*ImageNet. All the models are trained and get converged in 50 epochs. Note that each experiment is repeated for five times, and the averaged results are reported. In term of the analysis in Sec. 4.2, we prune the fully connected layer, Block #4.0, and #4.1. The results are shown in Tab. 5.

When no parameters are pruned, the R-P can achieve better results than our PiT algorithms. These results make sense, since the ridge penalty does not enforce the sparsity to the network[1]. However, with the increased ratio of parameters pruned, the performance of R-P gets degraded dramatically. In contrast, the results of our methods in PiT framework get decreased very slow. For example, when only 50% are saved in all the layers, the Top-5 accuracies are reduced by only 0% and 1.1% for L-P / GL-P, and S-P / GS-P respectively. Remarkably, if we only save 1.57% of original parameters on those layers, the S-P / GS-P can still is as high as 65.17, which is only 13.78% performance dropped. Again, note that all the methods have not done any fine-tuning step, and only been trained in one round. That means our S-P / GS-P can indeed select the most expressive weights or filters, and thus reduce the size of networks.

### 4.4 Discussion and Future Work

As the experiments shown in these three datasets, our PiT indeed can learn to prune networks without fine-tuning. We give some further discussion and highlight the potential future works,

1. In all our experiments, our L-P / GL-P, and S-P / GS-P are applied to, at most, four layers in one network. Theoretically, our PiT algorithms should be able to be directly applied to any layers of DNNs, since PiT only adds some sparse penalties in the loss functions. However, in practice, we found that the network training algorithm, *i.e.*, SGD in Alg. 3, is unstable, if we apply the sparse penalties more than four layers. It will take much more time and training epochs to get the networks converged.

2. Essentially, our PiT presents a feature selection algorithm, which can dynamically learn the importance of weights and filters in the learning process; mostly importantly, we donot need any fine-tuning step, which, we believe, will destroy values and properties of selected weights and filters. Therefore, it would be very interesting to analyze the statistical properties of selected features in each layer.

3. Theoretically, we can not guarantee the orthogonality of weights and filters in the trained model. Empirically, we adapt some strategies. For example, the weights and filters of each layer can be orthogonally initialized; and we apply the common regularization tricks, *e.g.*, dropout, and batch normalization. These can help decorrelate the learned parameters of the same layers. Practically, our PiT framework works well in selecting the important parameters and prune the networks as shown in the experiments. We also visualize the correlation between removed and none removed filters in the Appendix.

4. It is a conjecture that the capacity of DNNs may be too large to learn a small dataset; and it is essential to do network pruning. However, it is also an open question as how to numerically measure the capacity of DNNs and the complexity of one dataset.

---

[1]In practice, ridge regression may have better performance than lasso.

## 5  CONCLUSION

This paper proposes a Pruning in Training (PiT) framework. We add the sparse penalties in training the networks, and the weights and filters can be ranked via their learned parameter values. The networks can thus be directly pruned via the ranked weight order. Our framework can show good results on several deep learning benchmark datasets.

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

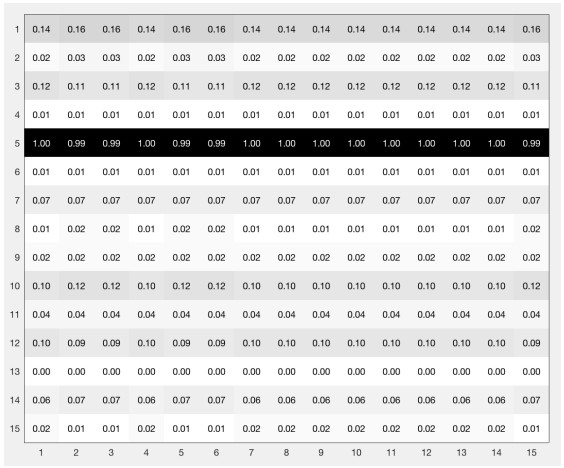

(a) Correlation between pruned and remaining filters

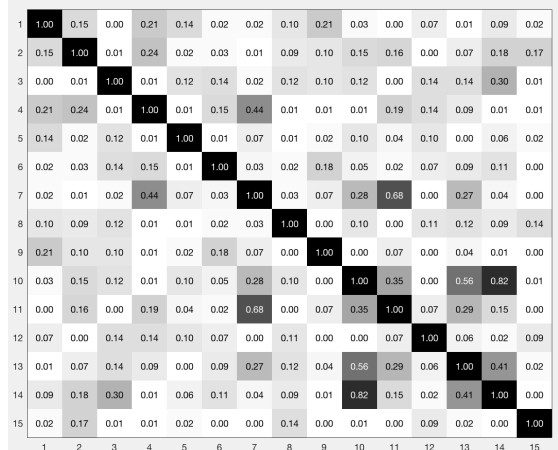

(b) Correlation between remaining filters

Figure 1: Correlation matrix of conv.c5 before and after the pruning. Note that we randomly select 15 filters to visualize in (a).

Bo Zhao, Xinwei Sun, Yanwei Fu, Yuan Yao, and Yizhou Wang. Msplit lbi: Realizing feature selection and dense estimation simultaneously in few-shot and zero-shot learning. *arXiv preprint arXiv:1806.04360*, 2018. 3.3, 3.4

Aojun Zhou, Anbang Yao, Yiwen Guo, Lin Xu, and Yurong Chen. Incremental network quantization: Towards lossless cnns with low-precision weights. *ICLR*, 2017. 2.2

Chenzhuo Zhu, Song Han, Huizi Mao, and William J Dally. Trained ternary quantization. *ICLR*, 2017. 2.2

Hui Zou and Trevor Hastie. Regularization and variable selection via the elastic net. *Journal of the Royal Statistical Society: Series B (Statistical Methodology)*, 67(2):301–320, 2005. 3.2

# 6 APPENDIX

We visualize the correlation matrix of conv.c5 of LeNet-5 on MNIST dataset, in Fig. 1. We randomly select 1000 images from the testing set of MNIST. In Fig. 1(a), each row is corresponding to one pruned filter, and each column is corresponding to the remaining filter. We find that most of them have lower correlation with one exception. In Fig. 1(b), we show the correlation between the remaining filters.

| Ratio | 100% | 25% | 12.5% | 6.25% | 3.13% | 1.57% |
|-------|------|-----|-------|-------|-------|-------|
| Plain | 99.17±0.02 | 60.87±14.00 | 29.65±4.89 | 20.82±3.32 | 20.82±3.32 | 20.82±3.32 |
| Rand | 99.11±0.01 | 43.72±5.74 | 30.07±7.48 | 18.35±4.58 | 24.12±5.75 | 22.62±6.43 |
| R-P | 99.12±0.04 | 61.05±7.46 | 46.91±6.36 | 30.34±6.82 | 30.34±6.82 | 30.34±6.82 |
| L-P | 99.05±0.04 | 74.29±1.06 | 47.28±12.12 | 28.58±3.22 | 28.58±3.22 | 28.58±3.22 |
| S-P | 99.00±0.01 | 85.09±5.76 | 32.58±2.65 | 22.88±6.89 | 22.88±6.89 | 22.88±6.89 |

(a) Pruning the conv.c3 layer

| Ratio | 100% | 25% | 12.5% | 6.25% | 3.13% | 1.57% |
|-------|------|-----|-------|-------|-------|-------|
| Plain | 99.12±0.02 | 80.46±5.46 | 62.61±9.05 | 45.49±0.97 | 32.34±1.53 | 21.30±4.83 |
| Rand | 99.19±0.01 | 62.23±10.12 | 37.71±4.34 | 23.58±6.96 | 18.58±4.70 | 14.36±3.27 |
| R-P | 99.16±0.06 | 75.47±8.11 | 60.31±5.19 | 37.97±2.99 | 26.11±2.13 | 18.11±1.05 |
| L-P | 98.95±0.04 | 98.95±0.04 | 90.29±1.30 | 60.37±5.45 | 32.91±3.35 | 20.31±1.10 |
| S-P | 98.97±0.07 | 98.96±0.08 | 98.67±0.15 | 68.27±11.22 | 42.10±7.83 | 24.95±8.92 |

(a) Pruning the conv.c5 layer

| Ratio | 100% | 25% | 12.5% | 6.25% | 3.13% | 1.57% |
|-------|------|-----|-------|-------|-------|-------|
| Plain | 99.10±0.02 | 96.73±1.05 | 95.65±1.76 | 89.60±3.49 | 78.40±5.48 | 64.17±6.93 |
| Rand | 99.09±0.01 | 91.56±3.89 | 71.05±6.08 | 51.92±8.87 | 33.65±6.17 | 29.91±8.49 |
| R-P | 99.13±0.05 | 96.39±0.48 | 95.31±0.79 | 91.29±3.46 | 82.75±5.59 | 68.35±6.29 |
| L-P | 99.10±0.03 | 98.89±0.05 | 98.89±0.05 | 98.89±0.05 | 98.89±0.05 | 98.89±0.05 |
| S-P | 99.13±0.03 | 98.73±0.10 | 98.61±0.15 | 98.23±0.40 | 96.75±0.88 | 92.53±3.17 |

(a) Pruning the fc.c6 layer

| Ratio | 100% | 25% | 12.5% | 6.25% | 3.13% | 1.57% |
|-------|------|-----|-------|-------|-------|-------|
| Plain | 99.11±0.06 | 98.36±0.25 | 94.52±2.08 | 78.15±7.25 | 68.72±15.72 | 47.35±10.88 |
| Rand | 99.15±0.02 | 66.29±10.47 | 50.76±8.21 | 34.61±8.66 | 24.05±5.86 | 19.78±7.93 |
| R-P | 99.13±0.08 | 98.62±0.23 | 96.50±1.13 | 84.04±13.42 | 67.08±15.75 | 56.34±2.33 |
| L-P | 99.10±0.09 | 99.11±0.08 | 99.11±0.08 | 99.11±0.08 | 99.09±0.10 | 96.47±1.91 |
| S-P | 99.03±0.05 | 99.00±0.04 | 99.00±0.05 | 99.01±0.05 | 99.03±0.04 | 95.41±3.89 |

(a) Pruning the fc.f7 layer

Table 6: Pruning one layer in LeNet-5 on MNIST dataset.

| Ratio | 100% | 25% | 12.5% | 6.25% | 3.13% | 1.57% |
|---|---|---|---|---|---|---|
| Plain | 92.96 | 28.91 | 29.21 | 18.09 | 16.48 | 11.76 |
| Rand | 93.58 | 17.04 | 15.14 | 15.38 | 14.38 | 13.97 |
| R-P | 93.75 | 82.35 | 55.53 | 31.01 | 25.18 | 21.19 |
| L-P | 93.54 | 93.53 | 93.30 | 92.61 | 91.29 | 89.20 |
| S-P | 93.17 | 93.23 | 93.27 | 93.30 | 93.26 | 93.27 |

(a) Pruning the Block#1.0

| Ratio | 100% | 25% | 12.5% | 6.25% | 3.13% | 1.57% |
|---|---|---|---|---|---|---|
| Plain | 93.44 | 37.32 | 20.60 | 13.94 | 15.93 | 18.84 |
| Rand | 92.90 | 25.63 | 30.82 | 29.31 | 33.40 | 34.33 |
| R-P | 93.95 | 64.03 | 46.21 | 30.97 | 28.50 | 27.67 |
| L-P | 93.57 | 93.57 | 93.57 | 93.57 | 93.57 | 93.57 |
| S-P | 93.52 | 93.53 | 93.53 | 93.49 | 93.51 | 93.49 |

(a) Pruning the Block#1.1

| Ratio | 100% | 25% | 12.5% | 6.25% | 3.13% | 1.57% |
|---|---|---|---|---|---|---|
| Plain | 93.42 | 41.69 | 18.36 | 24.32 | 27.18 | 27.28 |
| Rand | 93.44 | 14.39 | 13.42 | 12.60 | 12.41 | 11.43 |
| R-P | 93.35 | 54.27 | 32.17 | 28.09 | 32.83 | 32.51 |
| L-P | 93.60 | 93.62 | 93.58 | 93.57 | 93.59 | 93.61 |
| S-P | 93.27 | 93.18 | 93.26 | 93.26 | 93.26 | 93.24 |

(a) Pruning the Block#2.0

| Ratio | 100% | 25% | 12.5% | 6.25% | 3.13% | 1.57% |
|---|---|---|---|---|---|---|
| Plain | 93.48 | 49.77 | 40.31 | 33.81 | 37.05 | 36.75 |
| Rand | 93.50 | 31.60 | 35.15 | 35.75 | 37.92 | 37.40 |
| R-P | 93.78 | 64.74 | 51.29 | 45.81 | 47.68 | 49.65 |
| L-P | 93.50 | 93.50 | 93.50 | 93.50 | 93.51 | 93.51 |
| S-P | 93.50 | 93.16 | 93.27 | 93.28 | 93.32 | 93.29 |

(a) Pruning the Block#2.1

| Ratio | 100% | 25% | 12.5% | 6.25% | 3.13% | 1.57% |
|---|---|---|---|---|---|---|
| Plain | 93.41 | 55.48 | 22.94 | 21.70 | 30.32 | 35.36 |
| Rand | 93.11 | 37.42 | 36.66 | 34.00 | 36.42 | 37.10 |
| R-P | 93.61 | 74.24 | 57.27 | 41.14 | 37.44 | 38.32 |
| L-P | 93.66 | 93.66 | 93.67 | 93.68 | 93.66 | 93.61 |
| S-P | 93.59 | 93.63 | 93.48 | 93.46 | 93.45 | 93.42 |

(a) Pruning the Block#3.0

| Ratio | 100% | 25% | 12.5% | 6.25% | 3.13% | 1.57% |
|---|---|---|---|---|---|---|
| Plain | 93.11 | 51.73 | 34.17 | 21.04 | 19.12 | 18.86 |
| Rand | 93.84 | 46.47 | 46.62 | 44.79 | 40.30 | 43.55 |
| R-P | 93.73 | 54.31 | 67.49 | 54.03 | 42.84 | 56.04 |
| L-P | 93.54 | 93.54 | 93.54 | 93.54 | 93.54 | 93.54 |
| S-P | 93.82 | 93.46 | 93.33 | 93.29 | 93.25 | 93.26 |

(a) Pruning the Block#3.1

| Ratio | 100% | 25% | 12.5% | 6.25% | 3.13% | 1.57% |
|---|---|---|---|---|---|---|
| Plain | 93.08 | 82.70 | 54.18 | 45.27 | 39.31 | 38.29 |
| Rand | 93.44 | 48.88 | 40.89 | 38.45 | 36.97 | 37.38 |
| R-P | 93.55 | 88.63 | 69.91 | 47.08 | 49.69 | 47.40 |
| L-P | 93.65 | 93.65 | 93.65 | 93.65 | 93.66 | 93.68 |
| S-P | 93.61 | 93.50 | 93.45 | 93.38 | 93.44 | 93.41 |

(a) Pruning the Block#4.0

| Ratio | 100% | 25% | 12.5% | 6.25% | 3.13% | 1.57% |
|---|---|---|---|---|---|---|
| Plain | 93.37 | 90.01 | 85.29 | 79.24 | 77.75 | 75.53 |
| Rand | 93.57 | 82.86 | 79.16 | 77.26 | 76.65 | 76.73 |
| R-P | 93.63 | 88.20 | 87.16 | 79.41 | 75.00 | 70.77 |
| L-P | 93.79 | 93.79 | 93.79 | 93.80 | 93.77 | 93.77 |
| S-P | 93.61 | 93.93 | 93.83 | 93.84 | 93.93 | 93.89 |

(a) Pruning the Block#4.1

Table 7: Pruning one block in ResNet-18 on Cifar-10 dataset.