# OpenReview forum: "PRUNING IN TRAINING: LEARNING AND RANKING SPARSE CONNECTIONS IN DEEP CONVOLUTIONAL NETWORKS"
_ICLR.cc/2019/Conference_

### Official Review · AnonReviewer3 · 2018-11-03
**pruning in training using lasso and split LBI penalties. Not convincing experiments**

**Rating:** 4
**Confidence:** 5

**Review:**

This paper introduces an approach to pruning while training a network. This is interesting and experiments show interesting results in several datasets including ResNet18

Here are a few comments:

 - Pruning or regularization for compression is not new. Alvarez and Wen have used group lasso types as suggested in the paper and some others such as Alvarez and Salzmann (Compression aware training NIPS 2017) and Wen (Coordinating filters ICCV2017) have used low-rank type of while training. How is this different to those? They also do not need any sort of fine tuning and more importantly, they show this can scale to large networks and datasets.

- These last two works I mentioned promote redundancy, similarly to what is suggested in the paper. Would be good to get them cited and compared. Important from those is the training methodology to avoid relevant overheads. How is that happening in the current approach


- While I like the approach, would be nice to see how this scale.  All for methods above (and others related) do work on full imagenet to show performance.  For ResNet, cleaning the network is not really trivial (near the block), is that a limitation?
- Why limiting experiments to small networks and datasets? Time wise, how does this impact the training time?
- Why limiting the experiments to at most 4 layers?
- I am certainly not impressed by results on fully connected layers in MNIST. While the experiment is interesting does not seem to be of value as most networks do not have those layers anymore.

- Main properties of this approach are selecting right filters while training without compromising accuracy or needing fine tuning. While that is of interest, i do not see the difference with other related works (such as those I cited above)

- As there is enough space, I would like to see top-1 results for comprehensive comparison.

- I think tables need better captions for being self-contained. I do not really understand what i see in table 5 for instance.
- Droping 13% of top5 accuracy does not seem negligible, what is the purpose there? Would also be interesting to compare then with any other network with that performance.
- What about flops and forward time? Does this pruning strategy help there?

---

### Official Review · AnonReviewer1 · 2018-11-05
**Needs more formalism regarding the regularization path**

**Rating:** 5
**Confidence:** 4

**Review:**



==Major comments==

You need to better explain how the regularization path is obtained for your method. It is not clear to me at all why the iterates from lines 5-8 in Alg 2 provide a valid regularization path.

I am very confused by section 3.4. Is the pruning strategy introduced in this section specific to LBI? In other words, is there some property of LBI where the regularization path can be obtained by sorting the parameters by weight? It seems like it's not specific to LBI, since you use this pruning strategy for other models in your experiments. Is this the right baseline? Surely there are other sparsification strategies.


How/why did you select 5e-4 for lambda? You should have tuned for performance on a validation set. Also, you should have tuned separately for each of the baselines. There is no reason that they should all use the same lambda value.

Can you say anything about the suboptimality of the support sets obtained by your regularization paths vs. if you had trained things independently with different regularization penalties?

I am very concerned by this statement:
"However, in practice, we found that the network training algorithm, i.e., SGD in Alg. 3, is unstable, if we apply the sparse penalties more than four layers."
This is important. Do you have idea of why it is true? This instability seems like an obstacle for large-scale deployment.

Are your baselines state of the art? Is there anything discussed in the related work section that you should also be comparing against?


==Minor comments==
The difference between Alg 2 and 3 is mechanical and should be obvious to readers. I'd remove it, as the notation is complex and it doesn't add to the exposition quality. Instead, you should provide an algorithm box that explains how you postprocess W to obtain a sparse network.

Your citation format is incorrect. You should either have something along the lines of "foo, which does X, was introduced in author_name et al. (2010)" or "foo does X (author_name, 2010)." Perhaps you're using \citet instead of \citep in natbib.

Algorithm box 1 is not necessary. It is a very standard concept in machine learning.

On the left hand of (5), shouldn't Prox be subscripted by L instead of P?

---

### Official Review · AnonReviewer2 · 2018-11-06
**network pruning in training**

**Rating:** 5
**Confidence:** 4

**Review:**

This manuscript presents a method to prune deep neural networks while training. The main idea is to use some regularization to force some parameters to have small values, which will then be subject to pruning.
Overall, the proposed method is not very interesting. More importantly, the manuscript only lists the percentage of pruned parameters, but did not compare the actual running time before and after pruning.

---

### Meta-Review · Area_Chair1 · 2018-12-17
**not convincing experiments and lack of novelty**

**Confidence:** 5
**Recommendation:** Reject

**Metareview:**

This paper propose to obtain high pruning ratio by adding constraints to obtain small weights. Reviewers have a consensus on rejection due to not convincing experiments and lack of novelty.